# CStone: A *de novo* transcriptome assembler for short-read data that identifies non-chimeric contigs based on underlying graph structure

Raquel Linheiro, John Archer ◉ *

CIBIO/InBIO, Centro de Investigação em Biodiversidade e Recursos Genéticos, Universidade do Porto, Vairão, Portugal

* john.archer@cibio.up.pt

**Data Availability Statement:** All relevant data are within the manuscript and its Supporting Information files.

## Abstract

With the exponential growth of sequence information stored over the last decade, including that of *de novo* assembled contigs from RNA-Seq experiments, quantification of chimeric sequences has become essential when assembling read data. In transcriptomics, *de novo* assembled chimeras can closely resemble underlying transcripts, but patterns such as those seen between co-evolving sites, or mapped read counts, become obscured. We have created a de Bruijn based *de novo* assembler for RNA-Seq data that utilizes a classification system to describe the complexity of underlying graphs from which contigs are created. Each contig is labelled with one of three levels, indicating whether or not ambiguous paths exist. A by-product of this is information on the range of complexity of the underlying gene families present. As a demonstration of CStones ability to assemble high-quality contigs, and to label them in this manner, both simulated and real data were used. For simulated data, ten million read pairs were generated from cDNA libraries representing four species, *Drosophila melanogaster*, *Panthera pardus*, *Rattus norvegicus* and *Serinus canaria*. These were assembled using CStone, Trinity and rnaSPAdes; the latter two being high-quality, well established, *de novo* assemers. For real data, two RNA-Seq datasets, each consisting of ≈30 million read pairs, representing two adult *D. melanogaster* whole-body samples were used. The contigs that CStone produced were comparable in quality to those of Trinity and rnaSPAdes in terms of length, sequence identity of aligned regions and the range of cDNA transcripts represented, whilst providing additional information on chimerism. Here we describe the details of CStones assembly and classification process, and propose that similar classification systems can be incorporated into other *de novo* assembly tools. Within a related side study, we explore the effects that chimera's within reference sets have on the identification of differentially expression genes. CStone is available at: https://sourceforge.net/projects/cstone/.

**Funding:** This work was funded by National Funds through FCT (Fundação para a Ciência e a Tecnologia) and FEDER through the Operational Programme for Competitiveness Factors (COMPETE), via a project awarded to JA, under the references POCI-01-0145-FEDER-029115 and PTDC/BIA-EVL/29115/2017. RL's post doctoral position was supported by this project under POCI-01-0145-FEDER-029115. The website of the Portuguese Foundation for Science and Technology is: https://www.fct.pt. The funders had no role in study design, data collection and analysis, decision to publish, or preparation of the manuscript. The website of the Portuguese Foundation for Science and Technology is: https://www.fct.pt The funders had no role in study design, data collection and analysis, decision to publish, or preparation of the manuscript.

**Competing interests:** The authors have declared that no competing interests exist.

## Author summary

Within transcriptome reference sets, non-chimeric sequences are representations of transcribed genes, while artificially generated chimeric ones are mosaics of two or more pieces of DNA incorrectly pieced together. One area where such sets are utilized is in the quantification of gene expression patterns; where RNA-Seq reads are mapped to the sequences within, and subsequent count values reflect expression levels. Artificial chimeras can have a negative impact on count values by erroneously increasing variation in relation to the reads being mapped. Reference sets can be created from *de novo* assembled contigs, but chimeras can be introduced during the assembly process via the required traversal of graphs, representing gene families, constructed from the RNA-Seq data. Graph complexity determines how likely chimeras will arise. We have created CStone, a *de novo* assembler that utilizes a classification system to describe such complexity. Contigs created by CStone are labelled in a manner that indicates whether or not they are non-chimeric. This encourages contig dependent results to be presented with increased objectivity by maintaining the context of ambiguity associated with the assembly process. CStone has been tested extensively. Additionally, we have quantified the relationship between chimeras within reference sets and the identification of differentially expressed genes.

## Introduction

In the field of transcriptomics, awareness of chimeric sequences has been present for many years [1,2], but with the expansion of short-read sequencing technologies [3], and the associated exponential growth of sequence information stored [4], chimera quantification has become essential. Within transcriptome reference sets, such as the cDNA databases available from Ensembl representing various species [5], or those that are *de novo* assembled from short-read RNA-Seq data, non-chimeric sequences are direct representations of transcribed genes, while artificially generated chimeric ones are mosaics of two or more pieces of DNA incorrectly pieced together. The latter occurring during library preparation [6,7], or during the *de novo* assembly process [8,9], where there is a requirement to traverse paths across graphs constructed from read data that ranges in complexity depending on the nature of the gene families being represented [10–12]. Chimeras also occur at a genomic level during *de novo* assembly, such as when inferring haplotypes [13,14], but the causes, and consequences, at a genomic level are different [15–17]. In genomic assembly the aim is to reconstruct fewer large contigs that represent chromosomes [18,19]. Contigs produced by genomic assemblers are often utilized within the scope of population studies, in conjunction with mapping of whole genome read data, in order quantify and compare nucleotide variation or to annotate coding regions [20,21]. In transcriptomics, the goal is to quantify tens of thousands of expressed genes, and gene isoforms, that differ in length and expression pattern [12,22]. Differences such as these have lead to a distinction in how algorithms, and data structures, are optimized for either genomic or transcriptomic level assembly. In the absence of an available transcriptome reference, there are many RNA-Seq short-read assemblers available including, ABySS [23], Trinity [24], BinPacker [25] and rnaSPAdes [26]. Approaches, such as that implemented within the more recent Stringtie2 [27], that combine short-reads with the longer ones produced by single-molecule mRNA sequencing techniques [28], developed by companies including Pacific Biosciences and Oxford Nanopore Technology, have demonstrated high reliability; and are likely to greatly reduce chimera content once such data becomes routine [29–31]. In addition to *de novo* approaches, pipelines that combine genomic references with usage of

splice aligners, such as Tophat2 [32], HISAT2 [33] and SOAPsplice [34], in order to map RNA-Seq reads, estimate exons, splice-sites and subsequent transcripts, are also available [35,36]. The fundamental role that these tools play in RNA-Seq data analyse is reflected in the range of approaches developed as well as in the many reviews and benchmarking studies published [17,37–41].

Chimeric contigs can closely resemble expressed transcripts, but patterns such as those between co-evolving sites [42], remapped read counts [43,44] and polymorphisms [45,46] become obscured, and chimera presence has a poorly quantified impact on data analysis [41,47,48]. In relation to transcriptomics, included within data analysis is at times the characterization of gene differential expression patterns, where the primary signal utilized are the differences in mapped read counts, across datasets allocated to differing conditions, relative to sequences within a reference [49]. Increasing levels of variation between reference sequences and the reads being mapped decreases mapping accuracy [50], and artificially generated chimeras created during *de novo* assembly increase such variation by: (i) erroneously swapping parts of expressed transcripts with others, (ii) introducing sequencing variation at breakpoints within chimeric paths and (iii) over extension of contigs. It is therefore advantageous to know whether or not *de novo* assembled reference contigs have the possibility of being chimeric, so that care can be taken when finalizing conclusions following data analysis. A crucial part of *de novo* transcriptome assembly of short-read data is the arrangement of information present within reads into structures that represent full or partial gene families. These take the form of graphs, mostly de Bruijn [9,24], but may also be created from overlap consensus approaches [9,51]. In the de Bruijn based approach millions of fragments of specified length, termed kmers, are extracted from reads and used as nodes. Edges are placed where kmers match with the exception of one over-hanging nucleotide. The sought after outcome is a one-to-one relationship between gene families and graphs created [52]. Despite kmer efficiency at representing sequence data [53–56], the graphs ability to represent complete biological complexity has not been fully determined [57,58], but to date, the approach is state-of-the-art in dealing with the vast quantities of short-read RNA-Seq data produced. Complexity and size of the transcriptome [59], read coverage [60], gene expression levels [57] and sequencing error [61] are some of the factors that influence the number and nature of the graphs produced. It is the complexity of individual graphs, as represented by the number of possible start and end nodes, along with the number of internal junctions and cycles, that determine the extent of path choice during contig construction. Based on this we have developed a short-read RNA-Seq *de novo* assembler, CStone, that labels each contig created in accordance to one of three levels of complexity reflecting the nature of the graph from which it was derived: (i) a single start node and a single end node along with no internal junctions, (ii) a single start node and multiple end nodes along with internal junctions but no cycles, and (iii) a single, or multiple, start node(s) and a single, or multiple, end node(s) with or without internal cycles, i.e. everything else. Our classifications are no more than a description of the pre-existing structure of the de Bruijn based graphs. Paths from the first two cannot be chimeric, the first being a graph possessing a single path, while the second being one where each path has a unique end point and no alternative routes. For graphs of the third type, CStone, similar to other graph-based assemblers, uses the metric of read coverage to aid in path selection, but this does not guarantee complete non-chimerism, although high-quality representations of underlying transcripts are achieved.

Reference contigs labelled in this manner encourage dependent results to be presented with increased objectivity by maintaining the context of complexity, and ambiguity, present during construction. For example, the identification of a differentially expressed gene associated with a level (iii*)* contig can be considered more speculatively, whilst for a level (i) contig, more certainty can be assumed. Our approach is not solving the problem of *de novo* assembled

chimeras, but it is improving the interface between assembly software and result interpretation. Additionally, the approach, or similar ones, is readily implementable within any graph-based assembler. As a demonstration of CStones ability to assemble data we compare contigs produced by CStone to those produced by two well-established assemblers, Trinity [24], and rnaSPAdes [26], using both simulated data from four species, *Drosophila melanogaster* (fruit fly), *Panthera pardus* (leopard), *Rattus norvegicus* (brown rat) and *Serinus canaria* (canary), as well as real data obtained from a study on alternative splicing in *D. melanogaster* [62]. Our results indicate that all three assemblers perform well, and that the increased information that CStone adds on chimerism can be of value. Finally, to further highlight the poorly quantified issue of chimeric contigs, we demonstrate the effects of chimeric content within reference sets on the detection of differentially expressed genes using DESeq2 [49]; thus further highlighting the need for current assemblers to incorporate information on graph complexity into their outputs.

## Design and implementation

### (1) *De novo* assembly

**(1.1) Graph construction.**   Kmers of length 40 nt, along with frequency of occurrence, are extracted from reads and stored in descending order, Those of low complexity, where a single nucleotide type makes up more than 80%, are removed. Each remaining kmer is placed into a node data structure. Nodes whose kmers overlap by 39 identical nucleotides are merged into a composite node, the kmer of higher frequency being maintained. A default kmer length of 40 nt was chosen to minimize kmers being identical by chance, S1 Fig, and because 40 falls between the default kmer lengths of the already established graph based assemblers that we tested CStone against. This value can be altered by using the–k parameter. Edges are placed between nodes were kmers are identical with the exception of up to 5 overhanging nucleotides. Unconnected graphs, i.e. groups of connected nodes, are then extracted and stored (S2 Fig).

For each graph, local cycles between adjacent nodes are removed, while non-localized paths between junctions are maintained. This is done by merging pairs of siblings that have a valid connecting edge between them. During the merge process, all incoming and outgoing edges, as well as the kmer of higher frequency, are maintained. Edge validity is checked because as merges proceed some edges may begin to reflect distances that were larger than the initial edge connecting criteria. Following refinement, for a given graph (Fig 1), all nodes with a single connecting edge, i.e. those on the periphery, belong to set E; these either initiate or terminate paths. Nodes with more than two connecting edges, i.e. junctions, belong to set J. All other nodes belong to set I. The sum of the contents of E, J and I is equal to the total number of nodes on the network.

**(1.2) Graph classification and Contig creation.**   For each graph, in order to orientate paths, the node from set I with the highest kmer frequency, and no circumventing paths is selected. Two sets, E1 and E2, are then populated with nodes that represent the starts and ends of potential paths. To achieve this, nodes within set I are sorted in descending order of kmer frequency. This ensures that nodes on the top of the list are those through which the highest numbers, or the most expressed, transcripts pass; kmer frequency being derived from read coverage that reflects both expression and regions of shared identity between transcripts. Starting at the top, nodes are selected in turn (Fig 2, step i) and temporarily removed from the graph (Fig 2, step ii), along with all connecting edges. If two unconnected sub-graphs do not result, i.e. paths exist around the removed node, then the node, along with all its previous connecting edges, are placed back into the graph and the next node in the list is tested (Fig 2, step iii). If two unconnected sub-graphs do result, all external nodes from one of these are placed

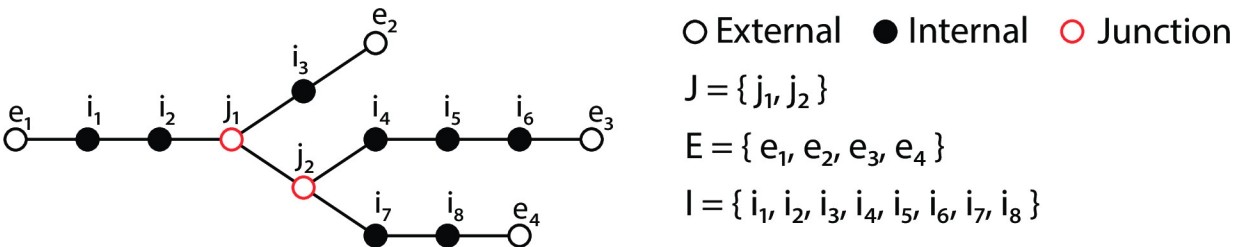

**Fig 1. An illustrative example of a single graph (of many) identified following edge connection.** Circular keys, top right, indicate node type, while black lines represent edges. The contents of each set, E, J and I, are shown.

into one set, and those of the other into a second (Fig 2, step iv). The smaller of these is labelled E1 and the larger E2. If they are the same size the choice of E1 and E2 labels is arbitrary. The removed node, along with its previous edges, is then put back and it is considered the cornerstone node of the graph, Fig 2, step v. CStone got its title based on this node.

Using paths that start in E1, end in E2, and that traverse the cornerstone node, we have defined three levels of classification, one of which will be associated with each graph: (i) no internal cycles, sets E1 and E2 each contain one node, resulting in a single non-chimeric path between E1 and E2 (Fig 3A), (ii) no internal cycles, set E1 contains one node, set E2 contains two or more nodes, resulting in a number of distinct paths equal to that of the number of nodes within E2, each of which is non-chimeric as no alternatives routes exist (Fig 3B) and (iii) all other cases, i.e. set E1 contains one or more nodes, set E2 contains one or more nodes and the graph may or may not contain internal cycles (Fig 3C). Cycles are identified within graphs by tracing all paths starting at E1 and identifying whether or not they can eventually double

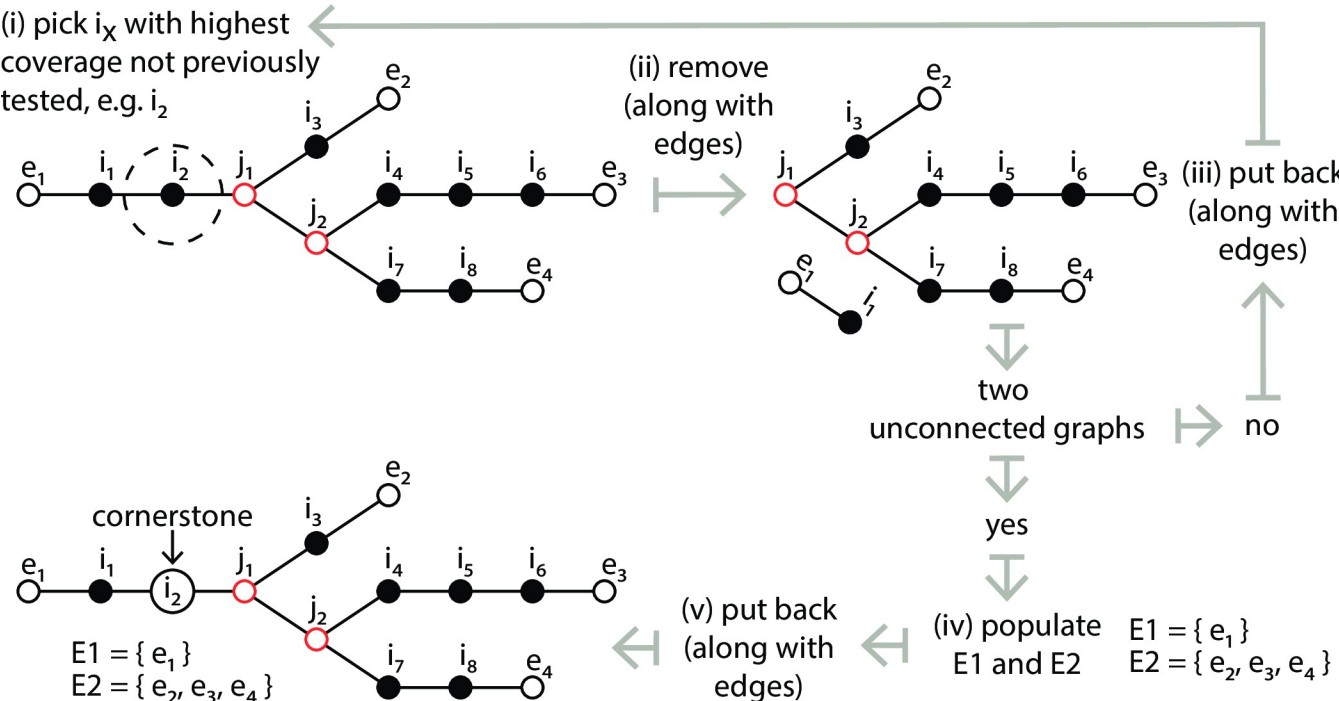

**Fig 2. Identification of the cornerstone node and population of sets E1 and E2.** Node indicators are the same as those used in Fig 1. Steps (i) to (v) outline the procedure to select the cornerstone node and subsequently to populate sets E1 and E2.

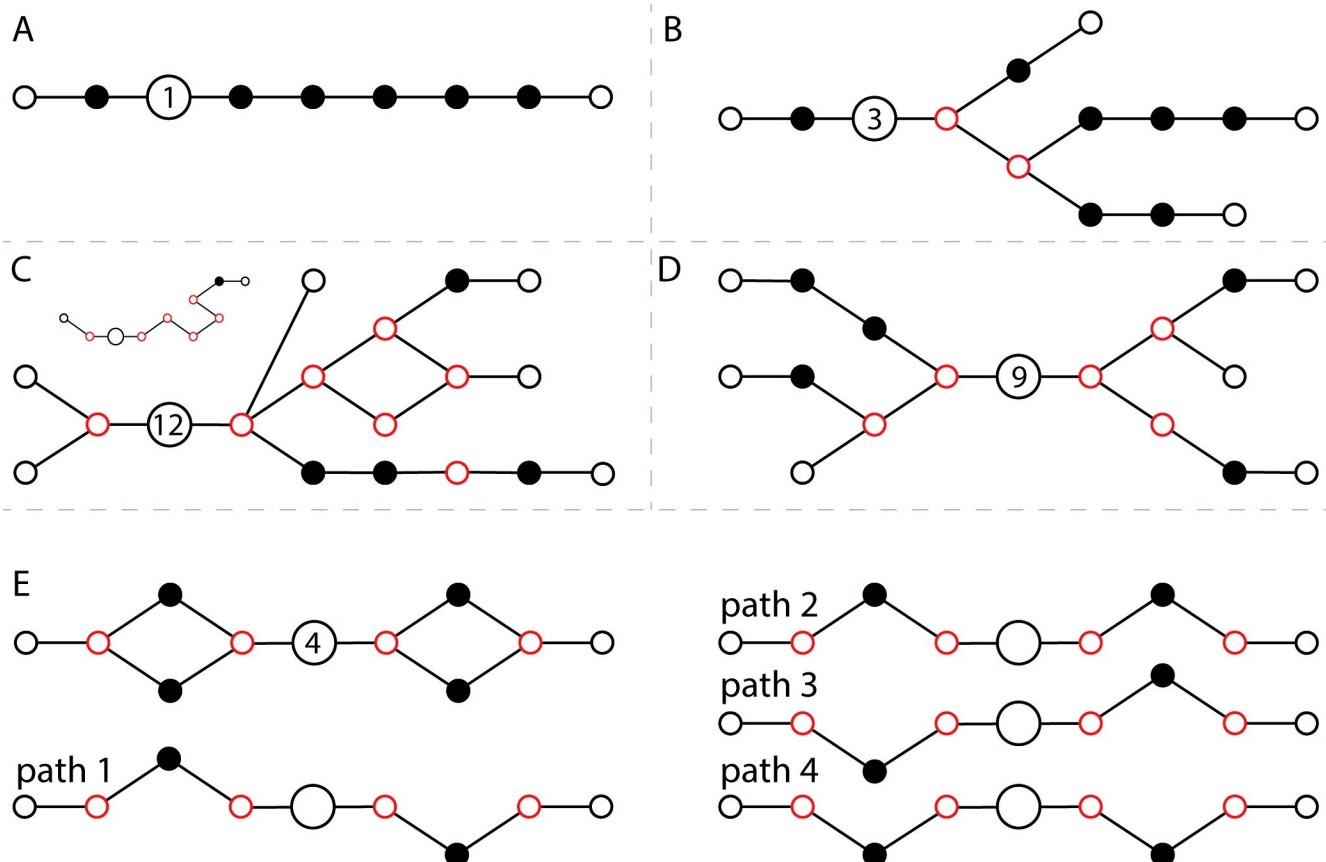

**Fig 3. Three levels of graph classification implemented within CStone.** Panels A to C display examples of graphs identified with classification levels 1 to 3 in order. The largest node indicates the cornerstone node whilst the number inside this indicates the number of possible paths passing through. In panel C one path (inset), of the possible 12, is shown. In the absence of read coverage information, as in this example, the path is most likely chimeric as it turns away from the nearest exit node and follows a more winding route. However, within real graphs coverage information could provide a justification for such a route. Panels D and (e) highlight two additional examples of graphs within classification level (iii). For the latter the four paths traversing the graph containing two cycles in sequence are shown, only two of which are required to reconstruct the original graph (paths 1 and 3) or (paths 2 and 4), leaving two possible chimeric ones.

back on themselves. Contigs are created in a manner that depends on the associated classification level. For level (i), a reconstruction of the kmers contained within each connected node on the single path is outputted as the contig. For levels (ii) and (iii), the first ten paths from each E1 starting node, level (ii) only having one node within E1, are sorted by mean read coverage and the top three are used to construct contigs in a similar manner to that done for level (i). The default value of three can be altered to a maximum value of five. All contigs produced are titled with a unique integer id as well as the graph classification level from which they were created. The latter is selected from LVL_1_NO_CYCLES_ONE_TO_ONE, LVL_2_NO_CYCLES_ONE_TO_MANY or LVL_3_COMPLEX.

Specifying the number of contigs per graph is necessary as it is not feasible to output all "possibly viable" paths from every graph. For some graphs there could be tens to hundreds of paths. We opted for outputting the top three best paths (by default), based on maximized coverage, in order to provide a reliable representation of gene families; in a way that could provide insight to contig non-chimerism. This strategy covers areas of analysis where obtaining reference sequences maintaining exact evolutionary relationships between sites is important, for example, when looking at co-evolving sites, geno-to-pheno altering polymorphisms or

recombinant-breakpoints. It also applies to differential experiments where the reliability of read counts at a gene family level out-weighs that of identifying ambiguous isoforms, many of which are artefacts of the short-read assembly graph traversal process. Limiting paths is less optimal if attempting to characterize all "true" isoforms from complex families; although given the advent of long-read sequencing technologies, the sole use of short-read data in conjunction with heuristic short-read *de novo* assemblers should be avoided. It should be noted here that the default top three paths per graph outputted by CStone are relative to each individual graph created. The manner in which graphs representing gene families are constructed is independent of read cover. As long as a gene family is represented by reads, a graph can be constructed. If on that graph there are many paths, it is the top thee, based on coverage, that are used. Importantly, this means that gene families associated with low expression will still be represented within the output.

## (2) Demonstration

**(2.1) Simulated data.** cDNA libraries from fruit fly, leopard, rat and canary, Table 1, were downloaded from Ensembl [5]. For each ≈10 million read pairs, of length 200 nt, insert size 300 nt and containing no read error, were generated from transcripts that ranged in length from 300 to 5000 nt using CSReadGen [63]. These datasets are available on the open-access repository Zendo and are associated with the url https://doi.org/10.5281/zenodo.5589533 [64]. The minor variation in final counts is due to reads being simulated in a manner that distributes them evenly over the reference transcripts, reflecting uniform coverage. Use of simulated data allows for the comparison of the assembled contigs to the sequences from which the reads were derived, while excluding the effects of unknown variation; including that of sequencing error and poor coverage. Additionally, with uniform coverage, for correctly assembled contigs the numbers of reads mapping back to them would be expected to correlate with length; reflecting that seen in S3 Fig.

For each species reads were assembled using CStone, Trinity (v2.12.0, kmer length of 25), and rnaSPAdes (v3.11.1, kmer length of 55). To assess assembly quality: (i) contig lengths were compared to cDNA reference transcripts. (ii) Bowtie2 [65] was used to map reads to each set of assembled contigs, after which read counts, obtained using the pileup.sh script of the bbmap package [66], were plotted against contig lengths. (iii) The megablast option [67], of the BLAST+ package [68], was used to identify how many of cDNA reference transcripts matched contigs produced by each assembler, as well as to assess the quality and the length of the matched regions. For each species-specific reference transcript the top 20 hits, within the contig file produced by each assembler, were examined. In each case, the matches producing the longest aligned regions were used to create plots of transcript length versus contig length, as well as contig length versus aligned region length. Plots were created using the R package [69]. A summary table of the percentage identities associated with the longest-match alignments,

**Table 1. Summary of simulated datasets.** The cDNA reference sequences (release-100) used as templates for simulation were downloaded from: https://www.ensembl.org/info/data/ftp/index.html.

| Species | Common name | No. of transcripts > = 300 && < = 5000 nt | Combined length of transcripts (nt) | No. of read pairs | Expected per site cover |
|---|---|---|---|---|---|
| *Drosophila melanogaster* | Fruit Fly | 26,680 | 56,006,763 | 9,986,804 | 71.33 |
| *Panthera pardus* | Leopard | 27,419 | 48,768,807 | 9,986,261 | 81.91 |
| *Rattus norvegicus* | Rat | 28,634 | 52,945,917 | 9,985,618 | 75.44 |
| *Serinus canaria* | Canary | 21,626 | 43,305,533 | 9,989,219 | 92.27 |

along with the number of unique reference transcripts finding a match, was also prepared. For CStone, species-specific bar charts were produced displaying the number of contigs generated from each of the three graph classification levels.

**(2.2) Real data.** Two adult fruit fly whole-body samples, from the Pang *et al.* (2021) study on alternative splicing [62], were downloaded from NCBI SRA, study no. SRP297872; run number SRR13251053 for adult 1 and run no. SRR13251054 for adult 2. Reads were 100 nt in length, and had been sequenced on Illumina's Hi-Seq 2000 sequencer. Following quality filtering, using Trimmomatic (LEADING:10 TRAILING:10 SLIDINGWINDOW:4:15 MINLEN:36 ILLUMINACLIP:TruSeq3-PE.fa:2:30:10) [70], they consisted of 31,543,384 and 29,812,987 read pairs. These were assembled using CStone, Trinity and rnaSPAdes, following which contig length distributions were summarized. Although these data were not generated directly from the fruit fly cDNA reference transcripts used in the previous section, it would be expected that, being representatives of the same species, the latter should align to many of the contigs assembled. For this reason megablast was used in a similar manner to that described for the simulated datasets.

## (3) Effects of chimerism on differential expression

To demonstrate the effects of chimerism within reference sets on downstream analysis, a differential expression experiment was repeated iteratively, on ten input read datasets, divided into two conditions, where during each iteration the proportion of chimeric reference sequences was increased (S1 Methods). All reference sets and corresponding datasets are available on the Zendo repository and are associated with the url https://doi.org/10.5281/zenodo.5589427 [71].

## Results

### (1) The software

CStone is written in Java and runs on operating systems with installed Java Runtime Environment 8.0 or higher. It is licensed under the GNU General Public License v3.0. At the time of writing, with under 2,000 lines of code, organized into 23 class files that result in an executable jar file of 72kb, it is minimalistic, clearly implemented and, if necessary, reproducible in a language of choice; for example within a learning environment. No external packages are required making setup or incorporation into other software projects, through inclusion of the jar file, relatively effortless. Table 2 indicates the assembly times required to assemble the datasets used within this manuscript.

**Table 2. Running times.** Times, in hours, minutes and seconds, taken by CStone to assemble datasets used in this study on Windows 10 running on 32 cores (AMD Ryzen Threadripper 2990WX @ 3.00GHz) and 128GB of ram, as well as on Ubuntu 20.04 running on 24 cores (Intel Xeon(R) CPU E5-2697 v2 @ 2.70GHz) and 64GB of ram. For simulated datasets "Effective transcriptome size" refers to the cDNA reference transcripts from which the reads were simulated, whilst for real data it is the (unknown) number of expressed genes within the adults that were sequenced.

| | Data Type | Effective Transcriptome Size | No Of Read Pairs | Windows 10 | Ubuntu 20.04 |
|---|---|---|---|---|---|
| **Fruit Fly** | simulated | 26,680 | 9,986,804 | 00:36:18 | 00:30:09 |
| **Leopard** | simulated | 27,419 | 9,986,261 | 01:01:34 | 00:51:02 |
| **Rat** | simulated | 28,634 | 9,985,618 | 01:05:14 | 00:19:07 |
| **Canary** | simulated | 21,626 | 9,989,219 | 00:37:57 | 00:31:36 |
| **Whole Adult 1** | real | N/A | 31,543,384 | 02:22:25 | 02:07:29 |
| **Whole Adult 2** | real | N/A | 29,812,987 | 02:14:19 | 01:52:03 |

**Table 3. Summary of the lengths of assembled contigs relative to the cDNA reference transcripts based on simulated data.**

| Species | Source | No. of contigs / transcripts* | Min. length | Median length | Mean length | Max. length | Outliers | Above 5000 nt |
|---|---|---|---|---|---|---|---|---|
| **Fruit Fly** | **Refs*** | **26680** | **301** | **1871** | **2099** | **5000** | **0** | **0** |
| | **CStone** | 20939 | 200 | 1138 | 1339 | 6658 | 581 | 6 |
| | **rnaSPAdes** | 19174 | 100 | 1034 | 1375 | 9126 | 729 | 126 |
| | **Trinity** | 24947 | 201 | 1574 | 1877 | 10806 | 341 | 464 |
| **Leopard** | **Refs*** | **27419** | **303** | **1494** | **1779** | **4999** | **235** | **0** |
| | **CStone** | 29778 | 200 | 1000 | 1218 | 6775 | 836 | 2 |
| | **rnaSPAdes** | 26603 | 100 | 970 | 1355 | 7467 | 587 | 21 |
| | **Trinity** | 33709 | 201 | 1285 | 1575 | 7918 | 627 | 113 |
| **Rat** | **Refs*** | **28634** | **301** | **1609** | **1849** | **5000** | **6** | **0** |
| | **CStone** | 26703 | 200 | 1188 | 1383 | 5276 | 572 | 3 |
| | **rnaSPAdes** | 25712 | 100 | 1272 | 1583 | 9036 | 117 | 72 |
| | **Trinity** | 36327 | 201 | 1204 | 1513 | 8844 | 843 | 219 |
| **Canary** | **Refs*** | **21626** | **303** | **1795** | **2002** | **4999** | **0** | **0** |
| | **CStone** | 21811 | 200 | 1067 | 1270 | 6638 | 556 | 2 |
| | **rnaSPAdes** | 24811 | 100 | 707 | 1104 | 8296 | 939 | 19 |
| | **Trinity** | 29399 | 201 | 1633 | 1889 | 8296 | 58 | 211 |

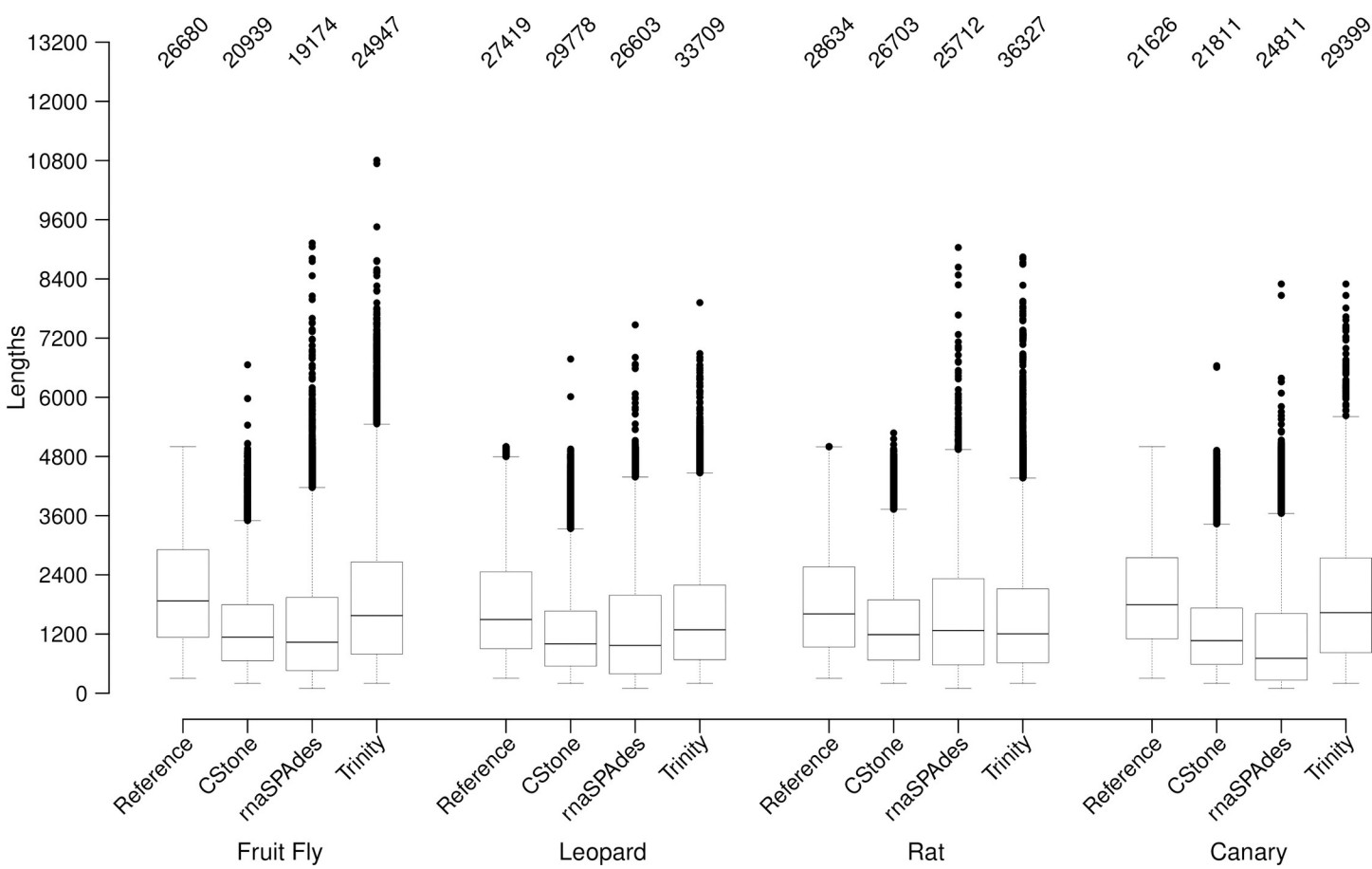

**Fig 4. Overall length of contigs and the cDNA reference transcripts from which the reads were simulated.** Dataset source along, with the species, is indicated along the x-axis. The numbers on the top indicate the total number of sequences present. Boxes represent the lengths falling within the inter quartile ranges. The median is shown within each box. Whiskers extend to the furthest data point that is within 1.5 times the inter quartile range and points beyond this are outliers (black circles). Outlier numbers are indicated in Table 3.

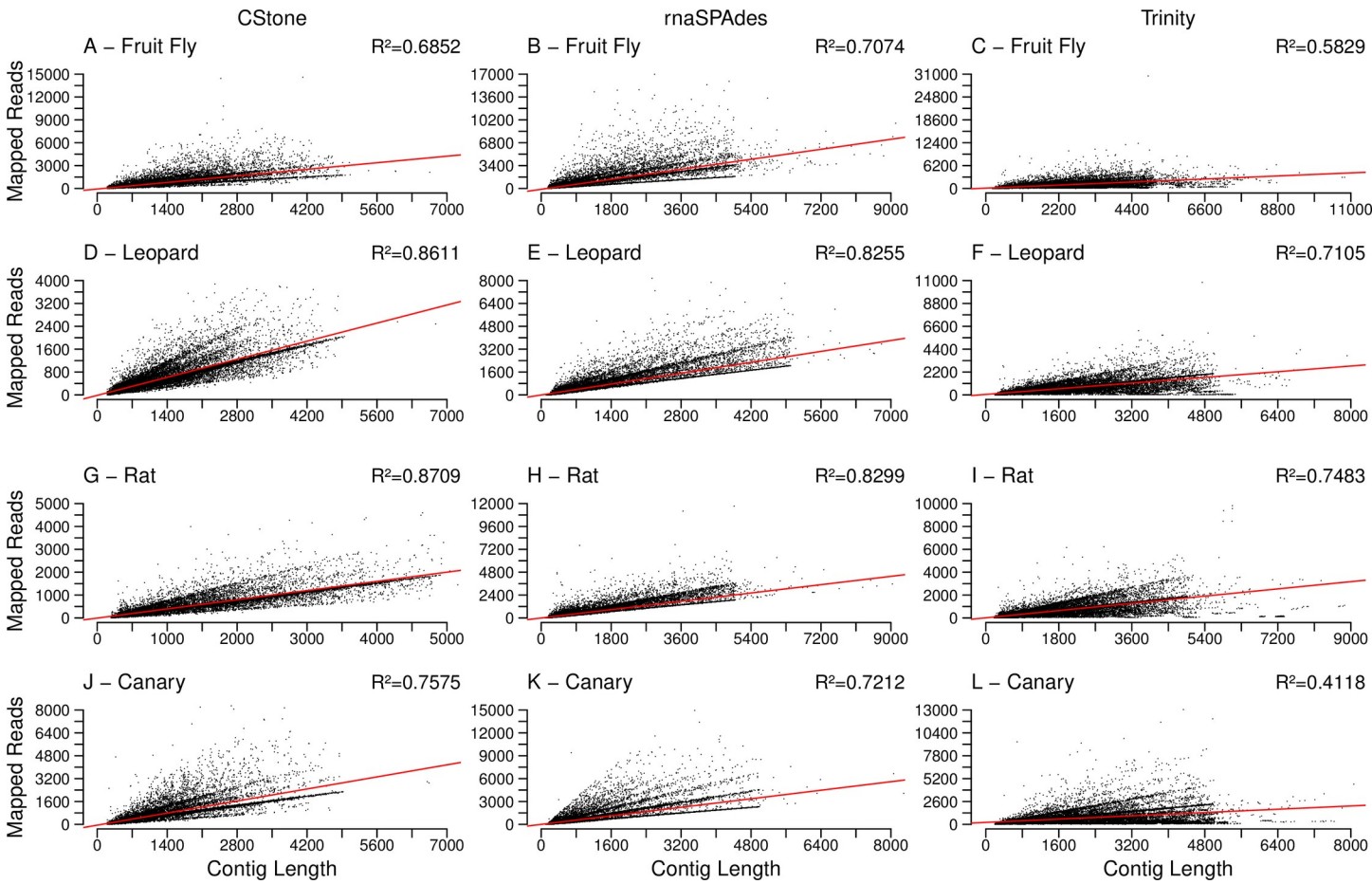

**Fig 5. Lengths of contigs versus the number of reads mapping to them.** The assembler used is indicated along the top of the figure, while the x-axis is labeled along the bottom. The red line indicates the line of best fit based on a linear model. $R^2$ values, located on the top right corners, indicate the correlation between mapped read counts and contig lengths, p-values of which are discussed in the text.

## (2) Demonstration

**(2.1) Simulated data.** Table 3 and Fig 4, compare the lengths of the contigs produced by each assembler to those of the cDNA reference transcripts. For each set of contigs the median length falls within the interquartile range of the reference transcripts. CStone produces some contigs beyond the length of the longest reference used, indicating some overextension, but the numbers of these are relatively low. For fruit fly, leopard, rat and canary, the overall numbers of contigs produced by CStone fall between those of rnaSPAdes and Trinity, the latter producing the highest numbers. In Table 3 it is observed that for CStone contig numbers were 20939 (fruit fly), 29778 (leopard), 26703 (rat) and 21811 (canary). These were produced from 18520, 29465, 25550 and 21517 underlying graphs respectively (S2 Fig). For Trinity the numbers of contigs created were 24947 (fruit fly), 33709 (leopard), 36327 (rat) and 29399 (canary), and were produced from 15136, 22181, 24077 and 16678 underlying graphs, as derived from the output contig files. These numbers indicate that CStone, although creating fewer contigs relative to Trinity, does not represent fewer networks, where networks are striving to have a one-to-one representation of gene families.

When the reads from each species are mapped against contigs, and the length of contig versus read count plotted, Fig 5, CStone achieves comparable $R^2$ values to those of both Trinity

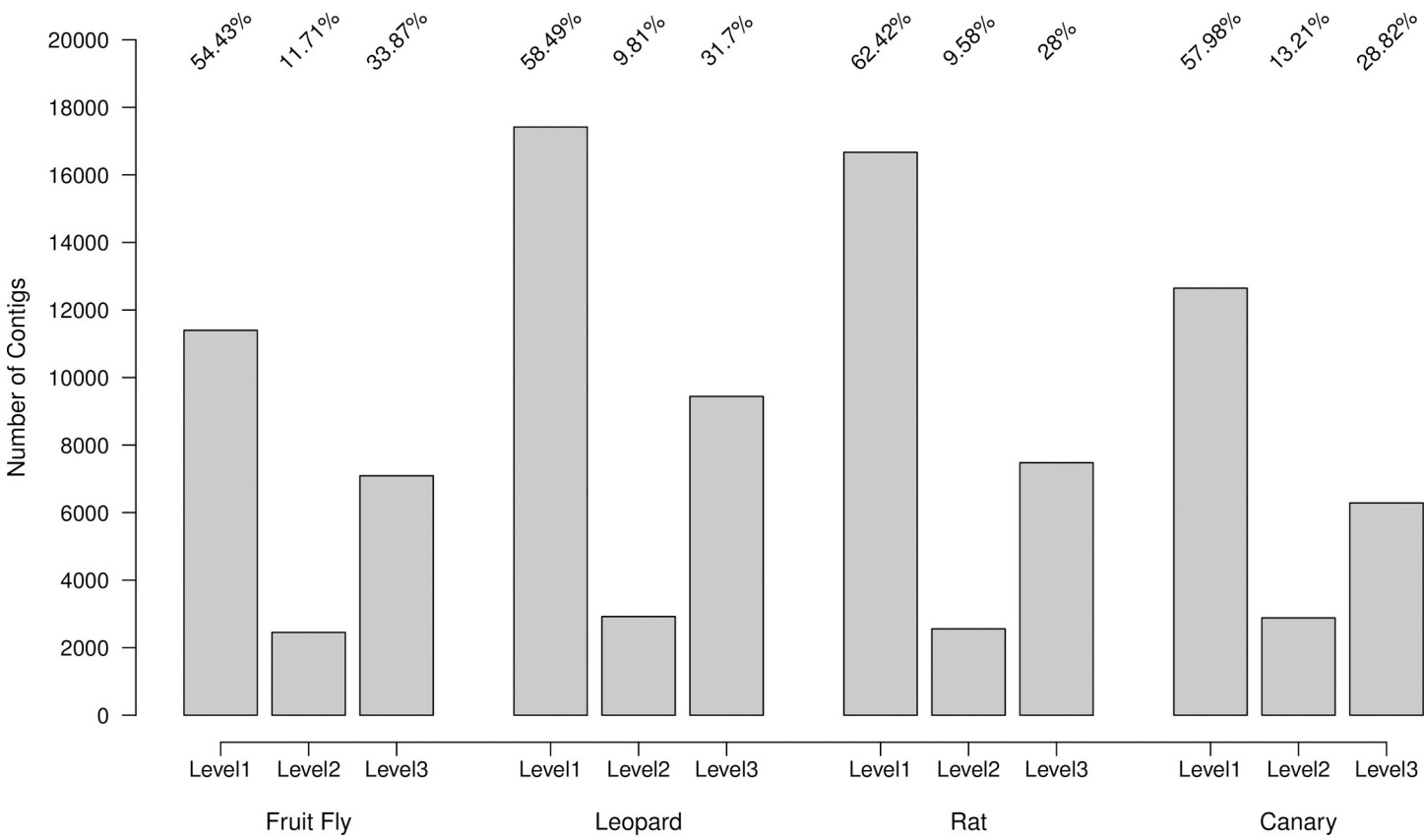

**Fig 6. CStone graph categories.** The numbers of contigs occurring within each of the CStone graph categories. Percentages of the total contigs are indicated above each bar.

and rnaSPAdes. For the Trinity assemblies of fruit fly and canary it is likely that a few contigs are lowering the $R^2$ value, for example, for fruit fly there is a single contig of length 4895 nt with 30,481 reads mapping to it that, when removed, increases the $R^2$ value from 0.5829 to 0.5944. In all cases, including that of the latter, there is a significant correlation, with all p-values below 2.2e-16, suggesting that the contig read counts are reflecting the nature in which the reads were simulated (S3 Fig).

For CStone the numbers of contigs associated with each of the three graph classification levels are displayed in Fig 6. Across the four species, for these datasets, an average of 58% and 11% of contigs come from graphs categorized as levels (i) and (ii). These are graphs that have structures that do not produce chimeric paths. Even when a contig from such graphs may not be complete, for example due to poor coverage, the proportion constructed is non-chimeric. To date, contigs produced by such graphs have been treated in an identical manner to those produced from the more ambiguous graphs classified as level (iii). CStone allows the user to make this distinction and discuss results related to such contigs in the context of the underlying graph complexity.

For each set of contigs, when the lengths of the reference transcripts are compared to the lengths of the best matching contigs, based on the longest aligned region as identified using megablast, Fig 7, a linear relationship is observed in all cases (p-values below 2.2e-16), indicating that reference transcript lengths are being reflected by the assembled contigs. For each contig, when the length of the aligned regions are compared to the contig length, (Fig 8), a significant correlation is also present in all cases (p-values below 2.2e-16), confirming that the

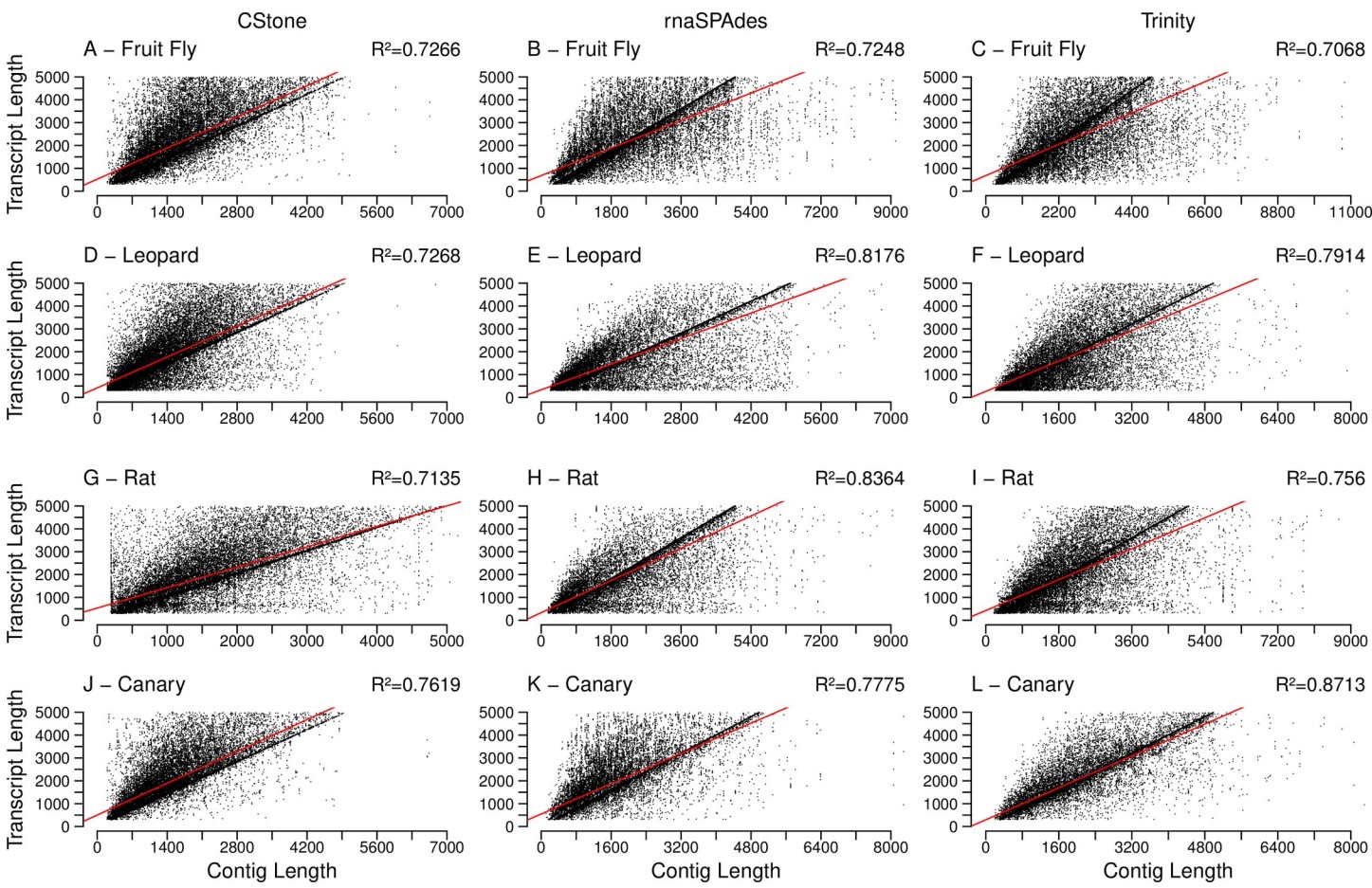

**Fig 7. Length of contigs versus length of reference transcript.** $R^2$ values, located on the top right corners, indicate the correlation between cDNA reference transcript lengths and contig lengths. The description of the rest of the figure is identical to that of Fig 5.

contigs are generally aligning over the majority of their lengths to the references to which they matched. The percent identities achieved within the aligned regions along with the number of different references being aligned to, are summarized in Table 4 and S4 Fig, and in both cases all values are high. Note: within S4 Fig although the range of identity values for CStone is generally wider, the means achieved for the four species are 99.80%, 99.25, 98.65 and 99.58%, and the four medians are 100%. Additionally, all values are above 70%.

The numbers of cDNA reference transcripts uniquely matching contigs produced by a single assembler, and those that match contigs produced by each of the different assemblers are presented in Fig 9. The majority of cDNA reference transcripts are represented by contigs produced by all three assemblers, indicating good agreement in overall transcriptome representation following assembly. Combined these results suggest that the contigs produced by the established assemblers Trinity and rnaSPAdes are of reasonable quality, and importantly, that those produced by CStone are of sufficient quality for demonstrating our approach to the inclusion of a graph-based metric indicating the extent of non-chimeric contig formation.

**(2.2) Real data.** Fig 10 and Table 5, summarize of the lengths of assembled contigs constructed from data derived from the two fruit fly whole-body samples. The mean contig lengths of rnaSPAdes and Trinity are higher, but the CStone median contig lengths fall between both the latter. For Trinity and rnaSPAdes the means are strongly influenced by the

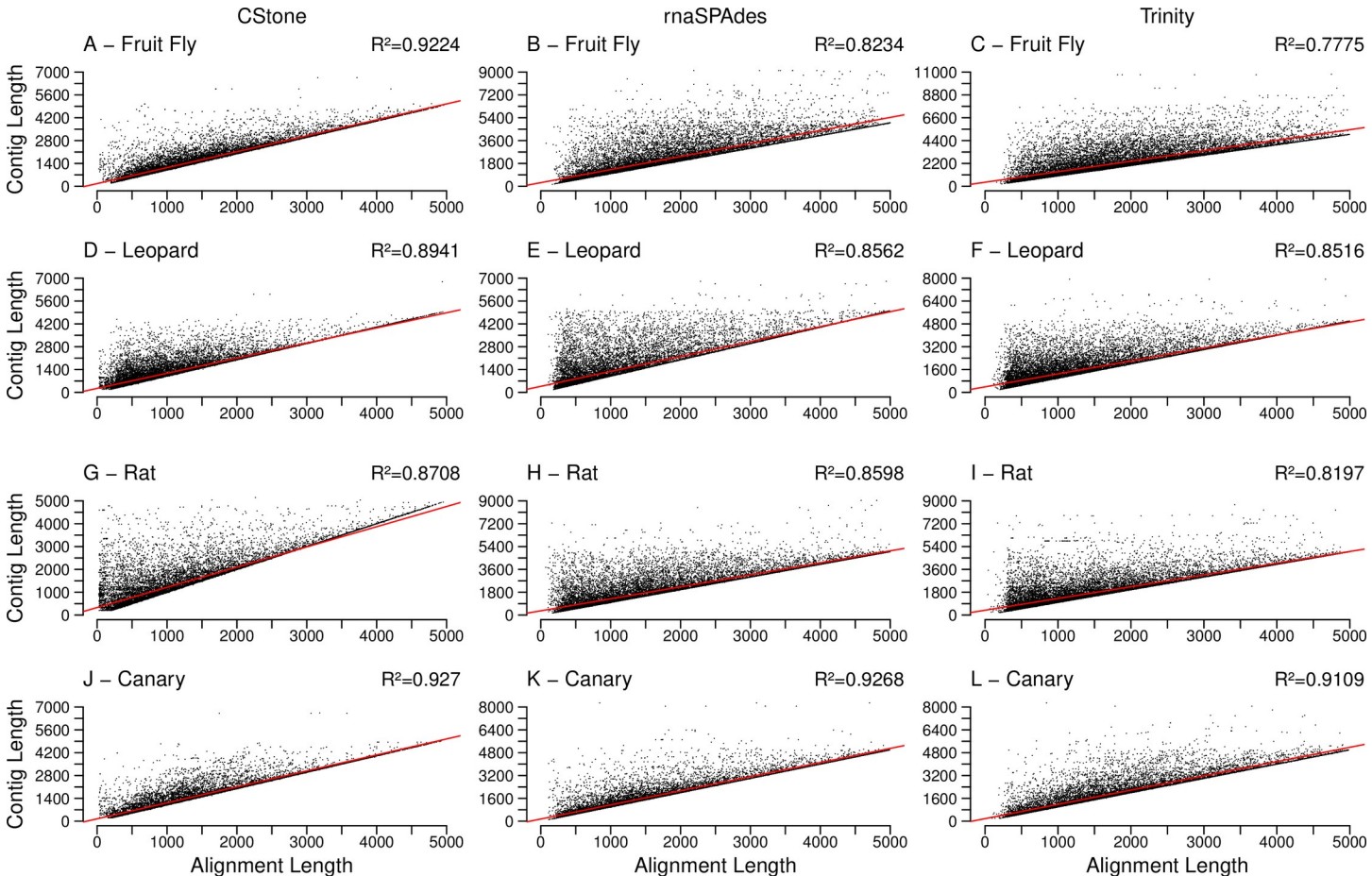

**Fig 8. Length of aligned region versus length of contig.** $R^2$ values, located on the top right corners, indicate the correlation between contig lengths and aligned region lengths. The description of the rest of the figure is identical to that of Fig 5.

**Table 4. Summary of percent identity between contigs and transcripts for the alignments represented in Fig 8.**

| Species | Source | Min. % identity | Median % identity | Mean % identity | Max. % identity | No. of cDNA transcripts matched | % of total cDNA transcripts |
|---|---|---|---|---|---|---|---|
| Fruit Fly | CStone | 74 | 100 | 99.80 | 100 | 26289 | 98.53 |
| | rnaSPAdes | 87 | 100 | 99.93 | 100 | 26678 | 99.99 |
| | Trinity | 81 | 100 | 99.90 | 100 | 26677 | 99.98 |
| Leopard | CStone | 70 | 100 | 99.25 | 100 | 26947 | 98.27 |
| | rnaSPAdes | 83 | 100 | 99.89 | 100 | 27418 | 99.99 |
| | Trinity | 76 | 100 | 99.71 | 100 | 27413 | 99.97 |
| Rat | CStone | 70 | 100 | 98.65 | 100 | 26585 | 92.84 |
| | rnaSPAdes | 81 | 100 | 99.75 | 100 | 28622 | 99.95 |
| | Trinity | 76 | 100 | 99.39 | 100 | 28603 | 99.89 |
| Canary | CStone | 71 | 100 | 99.58 | 100 | 21297 | 98.47 |
| | rnaSPAdes | 85 | 100 | 99.91 | 100 | 21625 | 99.99 |
| | Trinity | 83 | 100 | 99.82 | 100 | 21588 | 99.82 |

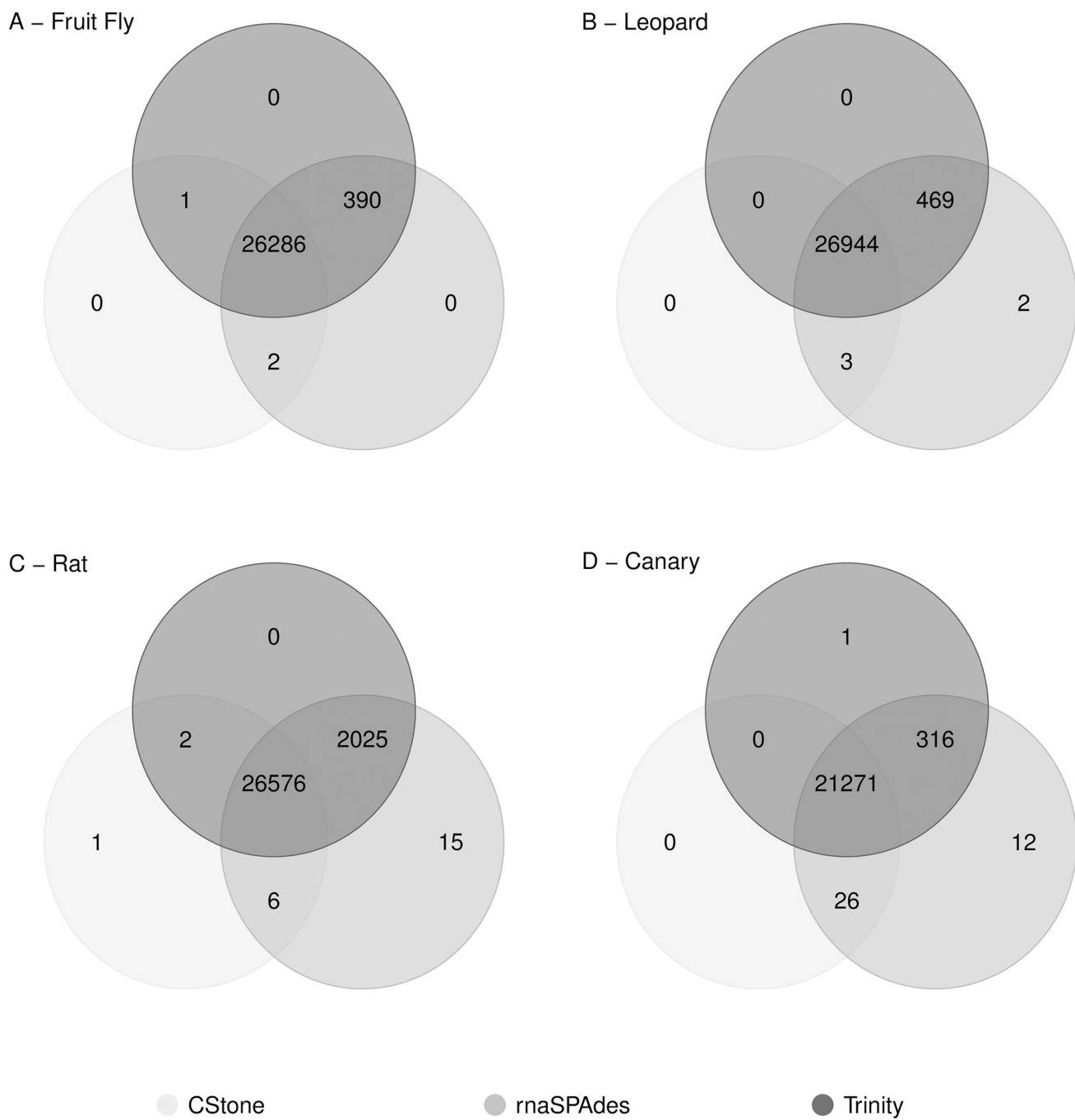

**Fig 9. Agreeability between assemblers.** Venn diagram showing the extent to which contigs produced by each assembler, when run on simulated data, agree in their representation of the species-specific cDNA reference transcripts. The key indicates the colour of the circle representing each assembler.

large number of transcripts above 5000 nt in length. Given that the lengths of transcribed genes are largely expected to be within the range of 300 to 5000 nt [72], such an increase in means, relative to the medians is more likely to be an indication of contig overextension rather than contig correctness.

The large portion of cDNA reference transcripts that matched contigs, Table 6, allowed for quantification of contig quality in terms of the length of matching regions versus over all

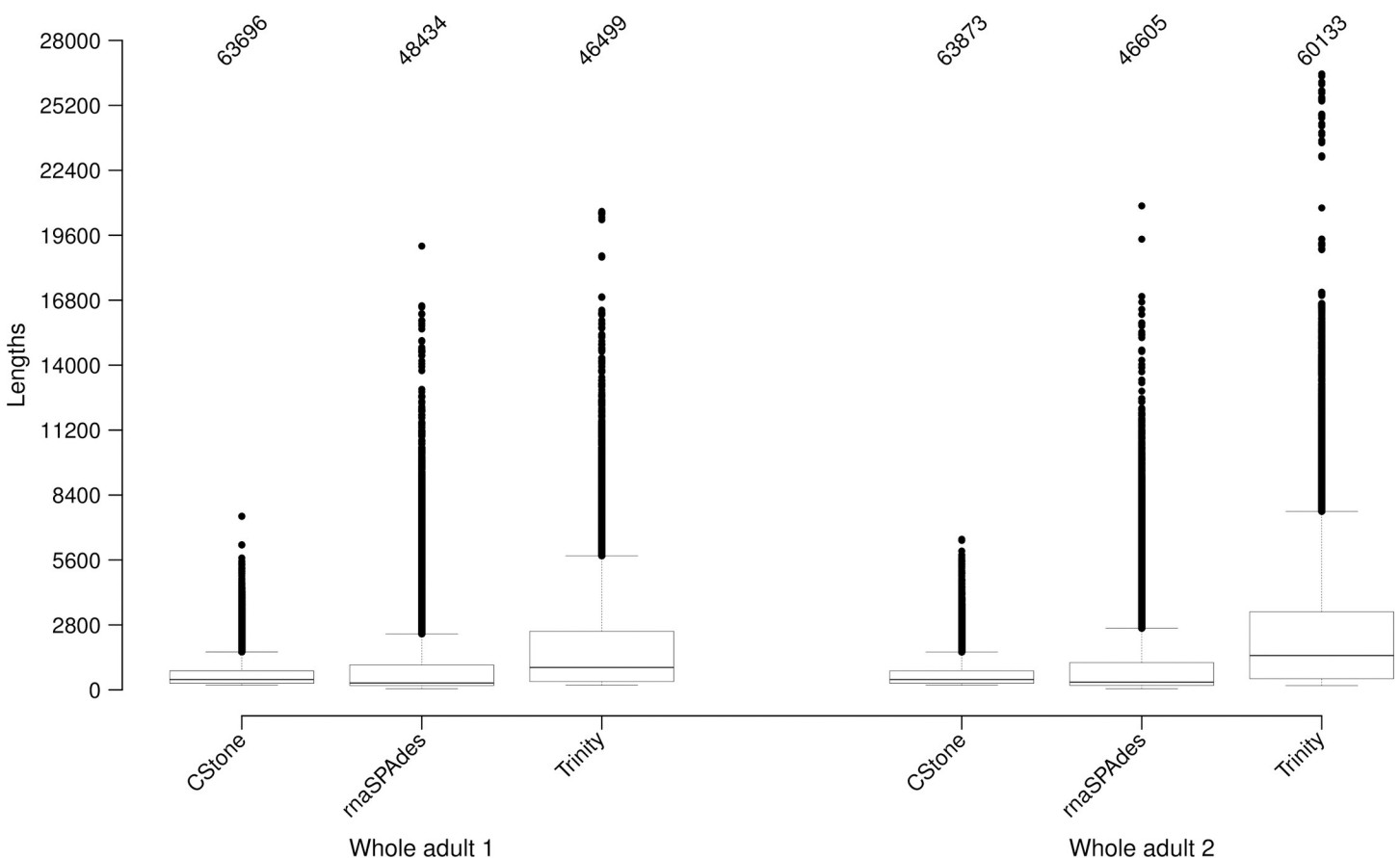

**Fig 10. Overall length of contigs produced on real data.** On the x-axis the dataset source along with the sample is identified. The numbers on the top indicate the total number of sequences present. Values covered by box and whiskers are the same as those described for Fig 4. The numbers of contigs above 5000 nt in length are indicated within Table 5.

contig length, Fig 11 and Table 6. CStone achieved notably strong correlations indicating assembly success, but the number of cDNA reference transcripts matched are on average 13% lower. This is likely due to the absence of overly large contigs above 5000 nt in length; where internal regions match many different reference transcripts. The largest contigs produced by CStone for whole-adult 1 and whole-adult 2 were 7,483 and 6,492 nt, while for Trinity and rnaSPAdes these numbers were 20,628 and 26,446 nt as well as 19,131 and 20,865 nt. CStone produced 19 such contigs for whole-adult 1 and 29 for whole-adult 2, while for Trinity the numbers were 3,666 and 7,742 and for rnaSPAdes they were 1,543 and 1,471.

**Table 5. Summary of the lengths of assembled contigs relative to the cDNA reference transcripts based on real data.**

| | Source | Min. length | Median length | Mean length | Max. length | Total no of Contigs | Above 5000 nt | % above 5000 |
|---|---|---|---|---|---|---|---|---|
| **Whole adult 1** | **CStone** | 200 | 444 | 652 | 7483 | 63,696 | 19 | 0.03 |
| | **rnaSPAdes** | 49 | 297 | 975 | 19131 | 48,434 | 1,543 | 3.19 |
| | **Trinity** | 201 | 967 | 1784 | 20628 | 46,499 | 3,666 | 7.88 |
| **Whole adult 2** | **CStone** | 200 | 446 | 654 | 6492 | 63,873 | 29 | 0.05 |
| | **rnaSPAdes** | 49 | 333 | 1016 | 20865 | 46,605 | 1,471 | 3.16 |
| | **Trinity** | 187 | 1482 | 2322 | 26556 | 60,133 | 7,642 | 12.71 |

**Table 6. Summary of percent identity between contigs and cDNA reference transcripts for the alignments represented in Fig 8 based on simulated data.**

| Species | Source | Min. % identity | Median % identity | Mean % identity | Max. % identity | No. of cDNA transcripts matched | % of total cDNA transcripts |
|---|---|---|---|---|---|---|---|
| **Whole adult 1** | CStone | 71.6670 | 99.7850 | 99.4143 | 100 | 21915 | 82.14 |
| | rnaSPAdes | 71.9070 | 99.6770 | 99.4088 | 100 | 25739 | 96.47 |
| | Trinity | 71.9070 | 99.6760 | 99.4120 | 100 | 25523 | 95.66 |
| **Whole adult 2** | CStone | 70.8950 | 99.7640 | 99.4033 | 100 | 22261 | 81.18 |
| | rnaSPAdes | 71.5970 | 99.6650 | 99.3990 | 100 | 25946 | 94.62 |
| | Trinity | 71.5970 | 99.6640 | 99.4050 | 100 | 25671 | 93.62 |

To investigate what proportion of contigs greater than 5000 nt in length were due to viral contamination, all viral reference genomes (≈10,000) from NCBI were downloaded (https://www.ncbi.nlm.nih.gov/labs/virus/). For whole-adult 1, out of the 1,543 and 3,666 contigs from rnaSPAdes and Trinity, a single match to a Nora virus genome, identified using megablast, was observed (S1 Table). Out of the 19 contigs from CStone no match occurred. For contigs below, or equal to, 5000 nt in length, each assembler produced just three matches, where the length of the matching region was above 200 nt. For CStone two of these being to the Nora

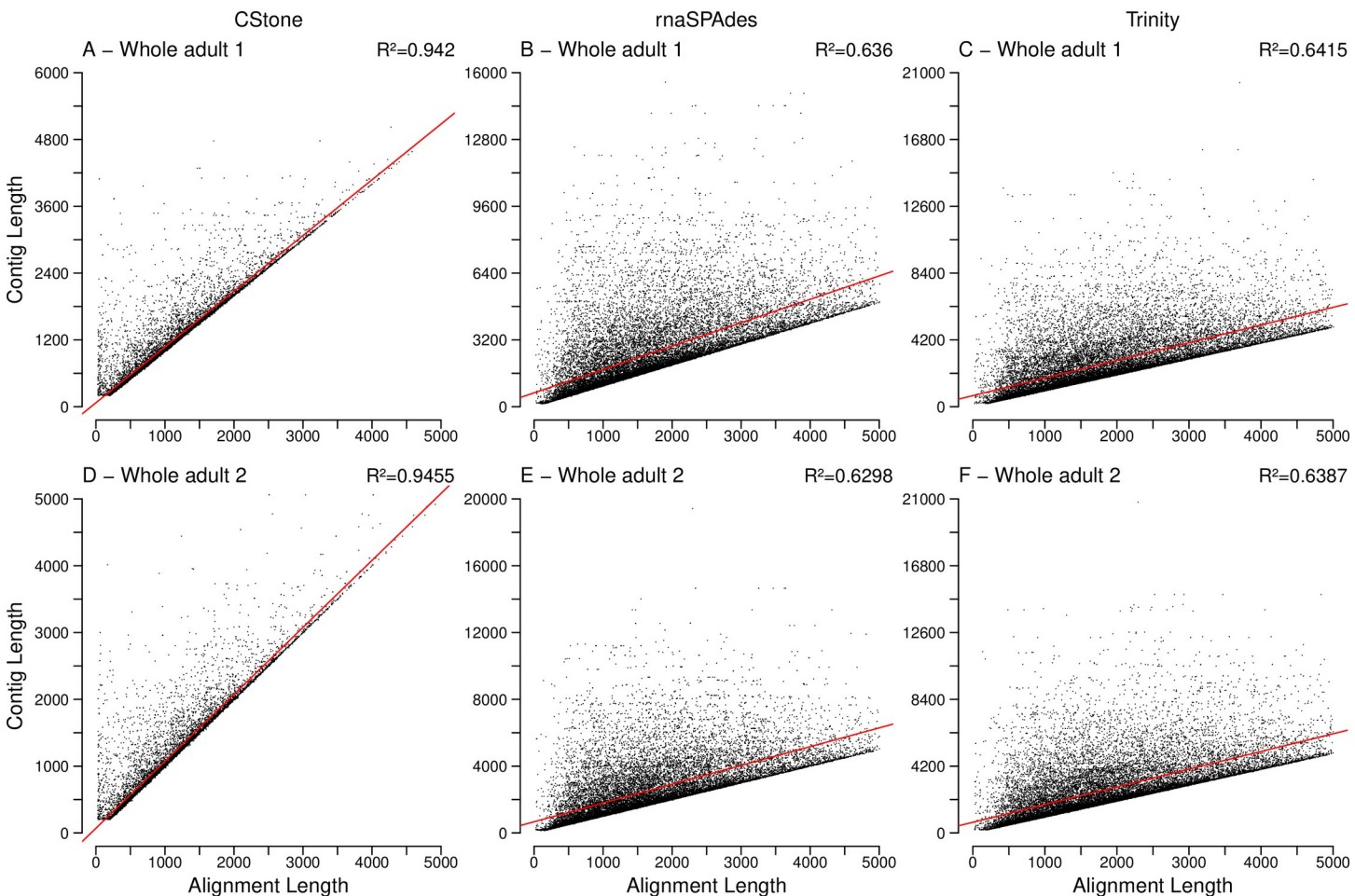

**Fig 11. Length of aligned region versus length of contig.** $R^2$ values, located on the top right corners, indicate the correlation between contig lengths and aligned region lengths. The description of the rest of the figure is identical to that of Fig 5.

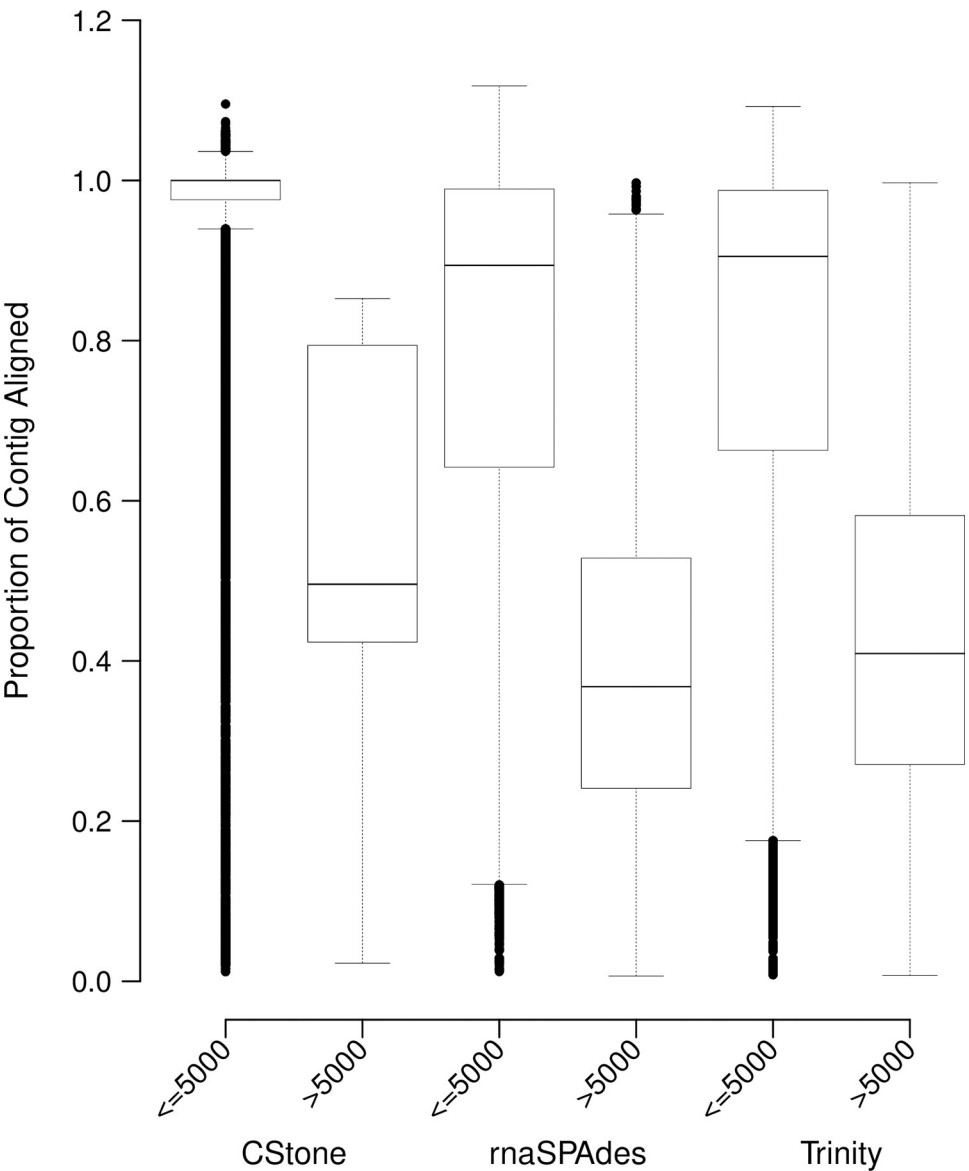

**Fig 12. Proportions of each contig aligned.** Boxes indicate the proportion of each contig aligned relative to its length. Values covered by box and whiskers are the same as those described for Fig 4.

virus previously identified. These results indicate that for whole-adult 1 contamination by virus genomes was minimum. For whole-adult 2 a similar outcome was seen (S2 Table).

For contigs greater than 5000 nt in length, the proportion of aligned regions, relative to contig length, are lower and have an increased range (Fig 12). This is likely due to contigs being overly extended relative to their best cDNA reference transcript match and/or having internal regions that do not align. Importantly for all assemblers, contigs below or equal to 5000 nt in length, produced far higher portions of aligned regions indicating completeness relative to matching cDNA references; those from CStone possessing the narrowest range of high values.

Similar to the simulated datasets, general agreement between the three assemblers for these data was high (Fig 13), although that between rnaSPAdes and CStone was highest; possibly due to the larger kmer sizes used by both the latter (S1 Fig).

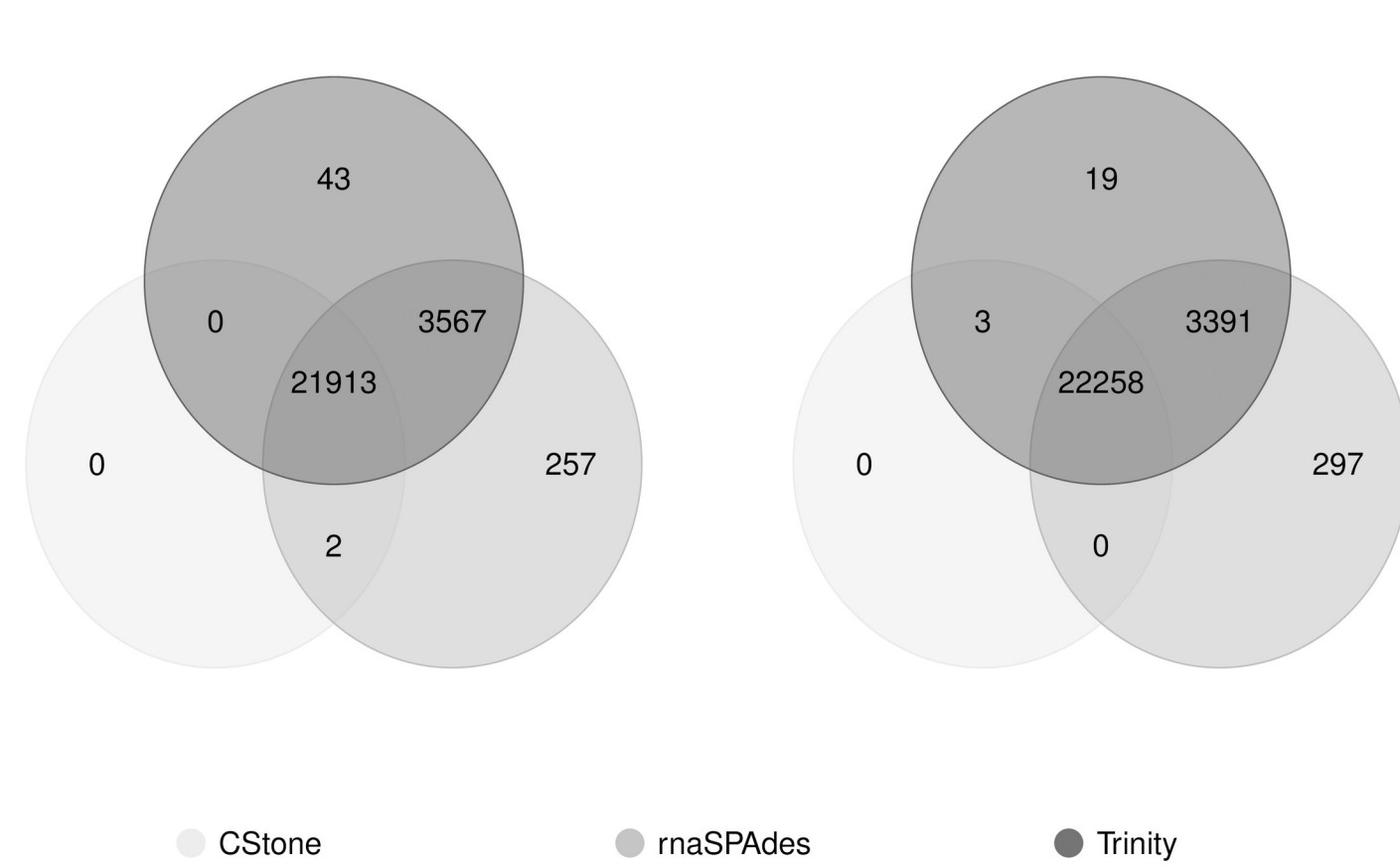

**Fig 13. Agreeability between assemblers.** Venn diagram showing the extent to which contigs produced by each assembler, when run on real data, agree in their representation of the species-specific cDNA reference transcripts. The key indicates the colour of the circle representing each assembler.

### (3) Effects of chimerism on differential expression

As the level of chimerism is increased within the reference set used, whilst the ten read datasets remain constant, the number of differentially expressed transcripts identified between conditions A and B varies (Fig 14); demonstrating that chimera presence is having an effect on their identification. Within the first Venn diagram, following just a 5% increase in chimeras relative to the non-chimeric reference set, there are 216 transcripts no longer detected as being differentially expressed (light grey), whilst there are 225 transcripts that are differentially expressed but that were not previously (dark grey). This trend continues up to the Venn diagram that compares the list of differentially expressed genes obtained using the 50% chimeric reference set. In this case there were many overly expressed transcripts due to the way in which background variation, and over expression, was applied. In non-simulated cases, where there is potential for few, to hundreds, of *de novo* assembled contigs being differentially expressed, it is important to be aware of the possibility of chimerism within each contig for two reasons. Firstly, individual chimeric contigs called as being differentially expressed are less than reliable having had their read counts altered erroneously during mapping and secondly, the presence of chimerism within the reference dataset as a whole has consequences for the count distributions used when calling differentially expressed contigs [49], whether those individual contigs are chimeric or not.

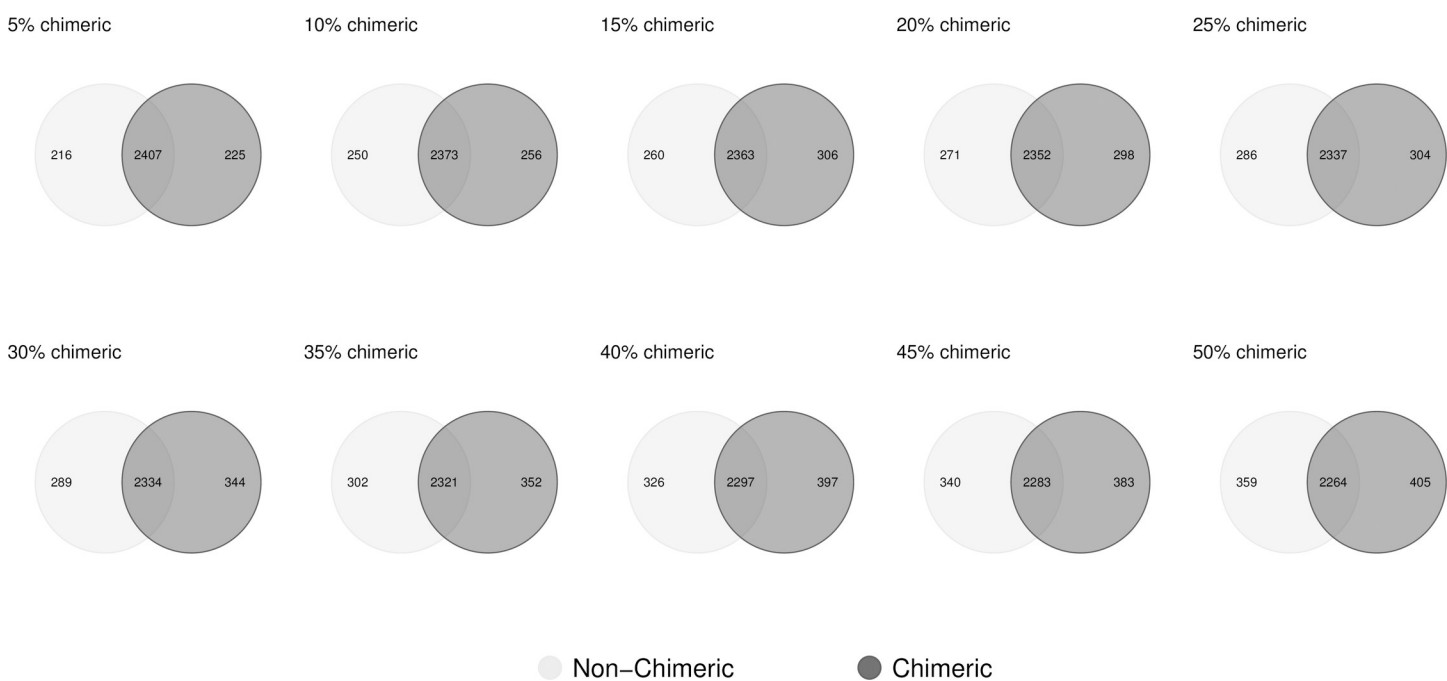

**Fig 14. Detection of differentially expressed genes in the presence of chimerism.** Light grey circles represent the number of identified differentially expressed genes, between the conditions A and B, that were detected in the absence of chimeric reference transcripts. Dark grey circles represent the number detected in the presence of chimerism, the extent of which is indicated as the percent value.

## Conclusion

CStone produces contigs of comparable quality to the two well-established assemblers that it was tested against. More importantly, it adds additional information to the output in relation to chimerism that: (i) can benefit the user, and research community as a whole, during the presentation and discussion of results, by maintaining the context of the ambiguities associated with chimerism when relevant and, (ii) is adaptable to the output of any *de novo* assembly tool implementing a graph-based approach. Additionally, we have demonstrated that the existence of chimeras within reference sets used for differential expression experiments has an effect on the detection of differentially expressed genes, thus highlighting the need to develop bioinformatics tools that aid in the quantification of such chimeras during *de novo* assembly.

## Availability

CStone is freely available, along with usage instructions, test data and source code, at the SourceForge project page: https://sourceforge.net/projects/cstone/. Simulated datasets used within our analysis are available on the open-access repository Zendo and are associated with the url's https://doi.org/10.5281/zenodo.5589533 [64] and https://doi.org/10.5281/zenodo.5589427 [71].

## Future directions

Areas of ongoing work include: (i) the incorporating of specialized data transformation and compression algorithms [73] into CStone in order to decrease assembly times and memory requirements. (ii) The sub-division of the level three graph classification category in order to associate each contig derived from such graphs with a likelihood score describing the extent of chimerism; such a score being dependent on the number of starting and end nodes as well as

the number and types of cycles present and (iii) on going maintenance and development of the tool to further enhance the quality of contigs produced based on user feedback. All developments will be released through the SourceForge project page.

## Supporting information

**S1 Fig. Unique kmer counts (y-axis) of specified length (x-axis) extracted from simulated reads.** Reads were simulated from the four species (indicated on right) as described under the "Demonstration" heading of the Design and Implementation section of the manuscript. Above kmer size of 18 little difference is observed in the number of unique kmers extracted. Kmer sizes from 1 to 18 show a marked increase as kmer frequency as size is incremented in steps of 1. This indicates that for these small kmers, shared kmers by chance (or kmer collisions) between different gene families and gene regions are more likely. For example, for kmer sizes of 4 there are only 256 unique permutations to describe the entire read dataset. Assemblies using kmers of this size would produce spurious sets of contigs that are highly chimeric. The three assemblers used in this study were Trinity, CStone and rnaSPAdes and have default kmer sizes of 25, 40 and 55 respectively, as indicated with the dashed vertical lines.
(TIF)

**S2 Fig. Graph sizes within CStone.** Following the edge connection step within CStone groups of connected edges, i.e. graphs, are extracted prior to the software identifying contigs. The size range of these networks (box and whiskers) and total numbers (top) are indicated for each of the simulated datasets (same as for S1 Fig) from the four species used within this study. Boxes represent the sizes falling within the inter quartile ranges. The median is shown within each box. Whiskers extend to the furthest data point that is within 1.5 times the inter quartile range and points beyond this are outliers (black circles).
(TIF)

**S3 Fig. Mapping reads back to the cDNA transcripts from which they were simulated.** Simulated reads containing no sequencing error, and distributed evenly across all transcripts, were mapped back to the cDNA transcripts from which they were generated in order to visualize the expected linear relationship between mapped read count and cDNA reference transcript length. Subsequent contigs assembled from these reads should also reflect this linear relationship, if not it is the first indication of poor quality assemblies.
(TIF)

**S4 Fig. Percent sequence identity within regions aligned between contigs and cDNA reference transcripts.** These plots are a visualization of the sequence identities presented in Table 6.
(TIF)

**S1 Table. Virus reference genomes from NCBI that matched with contigs representing whole-adult 1 using megablast.**
(DOCX)

**S2 Table. Virus reference genomes from NCBI that matched with contigs representing whole-adult 1 using megablast.**
(DOCX)

**S1 Method. Demonstrating the effects of chimerism on differential expression analysis.**
(DOCX)

## Author Contributions

**Conceptualization:** John Archer.

**Data curation:** Raquel Linheiro, John Archer.

**Formal analysis:** Raquel Linheiro, John Archer.

**Funding acquisition:** John Archer.

**Investigation:** Raquel Linheiro, John Archer.

**Methodology:** John Archer.

**Project administration:** John Archer.

**Resources:** John Archer.

**Software:** John Archer.

**Supervision:** John Archer.

**Validation:** Raquel Linheiro, John Archer.

**Visualization:** Raquel Linheiro, John Archer.

**Writing – original draft:** John Archer.

**Writing – review & editing:** Raquel Linheiro, John Archer.

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
