## [Decision Letter · Decision Letter 0]

4 Aug 2021

Dear Dr Archer,

Thank you very much for submitting your manuscript "CStone: A de novo assembler that identifies non-chimeric contig sequences based on underlying graph structure." for consideration at PLOS Computational Biology.

As with all papers reviewed by the journal, your manuscript was reviewed by members of the editorial board and by several independent reviewers. In light of the reviews (below this email), we would like to invite the resubmission of a significantly-revised version that takes into account the reviewers' comments.

The novelty of the method proposed in this manuscript is in the classification of the contigs produced so we ask the authors to make a convincing argument about the correctness of the classification as suggested by the reviewers.

We cannot make any decision about publication until we have seen the revised manuscript and your response to the reviewers' comments. Your revised manuscript is also likely to be sent to reviewers for further evaluation.

Sincerely,

Mihaela Pertea

Software Editor

PLOS Computational Biology

Mihaela Pertea

Software Editor

PLOS Computational Biology

Reviewer's Responses to Questions

**Comments to the Authors:**

Reviewer #1: Linheiro and John Archer presented a computational approach to de novo transcriptome assembly with an emphasis on identifying chimeric contigs. The concept is important for the field and the paper was written clearly. CStone seems to be easily implemented across operating systems, and can also be incorporated into existing tools. But the concept needs to be developed further to warrant publication in PLoS Computational Biology.

The primary concern with this tool is that ~30% of all contigs assembled were identified as 'chimeric,' which are claimed to affect downstream analysis such as differential gene expression. If this is indeed true and accurate, it would have large ramification for the entire field. If the authors could test out this concept further, how do these chimeric contigs affect differential expression, what's the scope of this issue, how can the identification of the chimeric contigs improve DE analysis, etc., it would help the reviewers to more accurately assess this approach and the conclusion that this tool can 'greatly benefit the user.'

Another issue that needs to be addressed is that especially when using real data, contigs generated by CStone have a wider spread towards shorter contigs, compared with Trinity and SPADES (Figure S4). Also, the contigs generated by CStone seem to have lower percent identity overall (Table 6). Could these factors increase the possibility of identifying chimeric contigs, which could be potential not chimeric? This would further cast doubt on the capability of CStone to improve downstream analysis.

Minor comments:

Adding line numbers will help the reviewers to navigate the manuscript much more easily.

This might be writing preference, but it would help to sometimes add a comma to aid the reading of the sentences. For example, at the beginning of the third paragraph in Design and Implementation, "For each graph, local cycles ..." would seem more readable. And there are many examples throughout the manuscript. Even if not at all places, some commas would still aid the reading of the sentences, which might otherwise be confusing.

Second paragraph of the Introduction, approach's -> approaches, effects -> affects

Reviewer #2: De novo transcriptome assembly of short RNA-Seq reads is still a challenging problem although tackled but not fully solved by various methods and tools over the last decade. One particular problem lies in the reconstruction of chimeric contigs representing falsely assembled transcripts. This can happen because reads are first reduced to shorter kmers of specific lengths and put into a graph structure that is then traversed to find connected paths finally representing contigs and thus transcripts. However, junctions in the graph structure can result in chimeric contigs that mix up different transcript sequences that biologically dont belong together.

Here, Raquel Linheiro and John Archer present CStone, a basic de novo assembly tool that utilizes kmers and a de Brujin-like graph structure to not only assemble contigs but to further classify them according to their chimeric potential. Thus, each contig gets categorized based on its original representation in the graph, helping to decide if the contig might be chimeric and thus needs further investigation or should be considered carefully in downstream analyses.

CStone, written in Java, is compared against two other state-of-the-art assemblers on simulated and real RNA-Seq data and shows similar performance but, more importantly, adds the functionality to evaluate the chimeric level of resulting contigs. I think that the presented method to categorize contigs according to their chimeric potential is an interesting and important feature that could be added also to other transcriptome (and genome?) assembly tools to further guide researchers on their way of interpreting assembled contig data.

The code together with example data and a manual is publicly available on Sourceforge and I was able to execute the tool on a MacBook Air.

Below, I have some comments and suggestions. The Figures are generally nice and help to understand the approach, however, the manuscript could need some English language editing to be more precise and clear.

[1] Title

… a de novo [transcriptome] assembler … ?

In general, you could be more precise about the transcriptomic field your tool is developed for. Also, your paper is about RNA-Seq data and transcriptome de novo assembly and not DNA and genome assembly. While one could assume that based on the abstract and introduction, that might be not always clear to the reader. E.g. consider changing sentences like “A crucial part of the de novo assembly process...” into “A crucial part of the transcriptome de novo assembly process...”

Actually, from your sourceforge page this is a good sentence to introduce the tool and its scope that I miss in the abstract/paper: “CStone is a [de novo] short-read assembler for RNA-Seq transcriptomic data based on a de Bruijn like approach.”

Besides, do you think that your tool can also detect and classify chimeric contigs from a DNA assembly graph? Maybe as an future extension?

[2] Language

Please check language carefully, there are several minor things such as

high quality → high-quality

whole body → whole-body

etc..

Please pay attention to commas.

“Over extension of contigs, a form of chimera, has also been shown to have a negative impact on differential expression studies”

suggestion →

“Overexpansion/Overextension of contigs, a form of chimera, has also been shown to negatively impact differential expression studies.”

I have the feeling that in general writing “overextension” as one word might be better - but I’m not a native speaker.

“ is the arrangement information present”

→ do you mean

“ is the arrangement of information present”

“state-of-the art” → “state-of-the-art”

“For graphs of the third type, CStone similarly to other graph based assemblers, uses the metric of read coverage to aid in path choice, but this does not guarantee complete nonchimerism, although high quality representations of underlying transcripts are achieved.”

→ suggestion (and as an example, I think your paper could benefit from some general language editing)

“For graphs of the third type, CStone, similar to other graph-based assemblers, uses the metric of read coverage to support path selection, but this does not guarantee complete chimera freedom, although high-quality representations of the underlying transcripts are achieved.”

“For each graph local cycles between close neighbouring nodes are removed whilst maintaining non-localized paths between junctions”

→

“For each graph, local cycles between adjacent nodes are removed, while non-localized paths between junctions (intersections?) are retained.”

“During merger...” ? During the merge process? What does “merger” mean? Please check the paragraph.

“ For a given graph following refinement, figure 1, on any partial path, before, after or between neighbouring junction nodes, with the exception of external nodes, each node will have two outgoing edges that link to its nearest neighbours, one parent (incoming) and one descendent (outgoing), depending on the direction of traversal.”

→ a complicated sentence, hard to follow. But your Figures are really nice and much easier to understand than the text description. Suggestion: reference your figures more often explaining the graph structure and classification process.

“ thus talking a small step” → “ thus taking a small step”

“ established assemblers could incorporated” → “ established assemblers can incorporate”

“CStone derived its title after this node.” → “CStone got its title based on this node.”

Figure 2 caption: please check the caption, the sentence is hard to follow/unclear language.

“Trimomatic” → “Trimmomatic”

“above 5000 in length” → “above 5000 nt in length”, in general add values

[3] Be more precise about DNA or RNA.

For example from the abstract: “For real data two datasets, each consisting of ≈30 million read pairs, representing two adult D. melanogaster whole body samples were used.” I guess you mean “whole-body transcriptome samples”?

[4]

“Where no reference exists reads can be assembled de novo to create one. “

This is not precise. When no genome reference exists you can also not use RNA-Seq reads to assemble one de novo. Or you mean “Where no transcriptome reference exists, ...” Or maybe better rephrase:

“Where no genome reference exists, RNA-Seq reads can be still assembled de novo to create transcripts. For this specific task, many assemblers are available including ...”

[5]

“In addition, hybrid approaches that rely on the existence of a reference genome ...”

I think “hybrid approach” is not a good term here because it’s mainly understood in the community when you combine short- and long-read data for assembly. Also, better than citing TopHat2 (or mapping tools in general) here you could mention tools such as Stringtie2 that actually perform reference-based transcript assembly.

[6] short vs long-read transcriptome assembly and impact on chimeric contigs

In the whole manuscript, you never mention that CStone was developed (I assume) for the assembly of short-read RNA-Seq data. I think that’s important having long-read technologies on the rise that might solve such chimeric problems due to short kmers quite easily (“no assembly required”). Of course, your method is still important because of the vast amount of short-read RNA-Seq data out there. Nevertheless, mentioning the impact of longer reads potentially representing full transcript isoforms could be mentioned as well.

[7] kmer length

You are using a fixed kmer size of 40 for CStone. Is it possible to use other kmer sizes? Did you test other sizes? Edit: I just read at Sourceforge that you actually can adjust the kmer size, so you could mention that briefly in the manuscript as well.

[8] Figure 3

“In panel (c) one path (most likely chimeric), of the possible 12, is shown (inset).”

On what is your assumption based that this path likely results in a chimeric contig? Because the number of possible paths traversing the cornerstone node is high?

[9] Reconstruction of lowly expressed transcripts

“For levels 2 and 3 the first 10 paths from each E1 starting node, level 2 only having one node within E1, are sorted by mean read coverage and the top three are used to construct contigs in a similar manner to that done for level 1.”

→ Does this mean that your approach might miss transcripts that are lowly expressed? E.g. a specific transcript isoform that is just rare and thus lower expressed as others?

[10]

“Reads were assembled using CStone, Trinity, and rnaSPAdes”

Parameters and version numbers for Trinity and rnaSPAdes are missing.

[11]

Windows and Ubuntu specs: I think it’s enough to have the detailed specifications written once (either in the text or Table 2 caption).

[12]

Can you somehow distinguish a) chimeric contigs due to mis-assembly and b) chimeric contigs that are actually derived from alternative splicing? For example, roughly one-third of your contigs from simulated data fall into category 3 that indicates chimeric contigs (Figure 6). Thus, if I get this information now from CStone I can still not tell if these are “real” transcripts or assembly artifacts, or? Or is the goal mainly to distinguish contigs that are clearly non-chimeric (categories 1 and 2)?

[13]

“while for Trinity and rnaSPAdes these numbers were 20,628 and 26,446 as well as 19,131 and 20,865 respectively, table 5”

→ Did you check these exceptionally long contigs for the real data set? Can they be annotated or are they even representing some contamination (virus, ...)?

[14]

“Additionally, CStone is small in size with minimum code and no dependencies, simplifying modification and incorporation into other bioinformatics tools, packages and pipelines”

→ I generally agree but I think CStone written in Java might not be able to be integrated that easily into existing tools that are mainly Python-based? At least, while Java developers will be happy about that others might not :) However, the method idea itself can surely be re-written as an extension for existing and well-established assembly tools. By saying that, I’m afraid that sourceforge might not be the best code base for the idea of presenting code that should be re-used and integrated/adjusted. Most developers are used to GitHub or GitLab where you can also easily write and open issues along with referencing the code and sending merge requests or forking the whole project. Actually, that would really allow re-using your codebase. I’m not so experienced with sourceforge but for me, it seems like the platform is not so developer-friendly. Thus, you might want to consider moving your code to Git{Hub,Lab}? Don’t get me wrong, you already did a great job in documenting your code and writing manuals but I have the feeling your work could benefit from such a transition.

[15]

Sourceforge manual: “Each time the software is run, this directory will be deleated and recreated, so it is best to copy it somewhere else (or rename it) between usages. This was done to protect the user from accidentally deleting directories.”

→ I dont agree that this is good behavior. It would be better if the output directory gets a timestamp or unique name assigned or the user is warned if the output directory already exists instead of just deleting results the user might run hours or days for.

**Have the authors made all data and (if applicable) computational code underlying the findings in their manuscript fully available?**

Reviewer #1: Yes

Reviewer #2: Yes

PLOS authors have the option to publish the peer review history of their article (what does this mean?). If published, this will include your full peer review and any attached files.

Reviewer #1: No

Reviewer #2: **Yes: **Martin Hölzer
---

## [Decision Letter · Decision Letter 1]

19 Oct 2021

Dear Dr Archer,

Thank you very much for submitting your manuscript "CStone: A de novo transcriptome assembler for short-read data that identifies non-chimeric contigs based on underlying graph structure." for consideration at PLOS Computational Biology. As with all papers reviewed by the journal, your manuscript was reviewed by members of the editorial board and by several independent reviewers. The reviewers appreciated the attention to an important topic. Based on the reviews, we are likely to accept this manuscript for publication, providing that you modify the manuscript according to the review recommendations.

Sincerely,

Mihaela Pertea

Software Editor

PLOS Computational Biology

Mihaela Pertea

Software Editor

PLOS Computational Biology

[LINK]

Reviewer's Responses to Questions

**Comments to the Authors:**

Reviewer #1: The authors have addressed my comments and the revised manuscript is deemed adequate for publication.

Reviewer #2: Dear authors,

thank you very much for carefully answering my questions and especially for the additional analyses and effort you put into the manuscript. I think the manuscript significantly improved from the first version, not at least due to the language editing and the more precise wording (RNA-Seq, ...). Good job!

I'm also ok with your code living on SourceForge. However, the point that GitHub does not have enough space is not entirely valid for me. Someone that wants to use your code does anyway not expect to download 2GB of data from a repository. Thus, my preferred way would be to put code and light-weight content on a code repository and example data/large files on another resource such as Zenodo or OSF (Open Science Framework). Then, such data resources could be referenced from the code repository. However, I also agree that this is a matter of taste and you justified why you want to keep your code on SourceForge and I'm fine with that.

Also, thanks for the detailed explanation regarding the application for DNA graphs. You're right and after your detailed explanation, I think it is good that you clearly focus on this chimeric problem in the context of short-read RNA-Seq data.

Regarding

[9] Reconstruction of lowly expressed transcript

Thanks for the explanation, this makes it more clear. I also better understand now why you are limiting the number of contigs derived from the graph and thus focusing on the paths with higher coverage as a priority. However, this still implies that low-coverage transcripts might be just missed by your approach, or? Did you look into that? E.g. in your simulated data did you also had low-coverage contigs and how do rnaSPAdes, Trinity, and CStone perform then? I dont want to produce too much extra work here that might be out of the main scope of the paper, especially bc/ you nicely wrap the topic up via:

"Here we are effectively saying that we are not solving thegeneral short-read assembler problem, but ratherwe arehighlighting it and attempting to give the user some intuitive access to the contigs in the context of theproblem;while at the same timeoutputting what we believe are the most reliable contigs from each family present."

in your p2p letter. Thus, I'm also fine with a discussion of this topic and making the reader/user aware of this, if further needed.

Besides, this very long and encapsulated paragraph you newly proposed could be simplified bc/ it is hard to follow:

"This covers many bioinformatic needs, including: (i) that of obtaining reliable references from gene families,where information on chimerism is relevant, such as when looking at co-evolvingsites, geno-to-pheno alteringpolymorphisms or recombinant breakpoint and (ii) many differential expression experiments where the advantage of having information on the reliability of read counts, see method S1 from subsection (3), out-weighs that of obtaining large lists of differentially expressed transcripts based on annotating and mapping to potentially hundreds of thousands of poorly supported contigs."

Also, I think:

" ... or recombinant breakpoint[s]..."

and

"... long-read sequencing " (missing the dash)

also

"... short-read data would be a poor choice for this"  maybe better "... which are anyway difficult to fully reconstruct from short reads alone" (or something like that)

Thank you very much for all the work

**Have the authors made all data and (if applicable) computational code underlying the findings in their manuscript fully available?**

Reviewer #1: Yes

Reviewer #2: Yes

PLOS authors have the option to publish the peer review history of their article (what does this mean?). If published, this will include your full peer review and any attached files.

Reviewer #1: No

Reviewer #2: **Yes: **Martin Hölzer

Figure Files:

Data Requirements:

Reproducibility:

References:

---

## [Editor Report · Decision Letter 2]

11 Nov 2021

Dear Dr Archer,

We are pleased to inform you that your manuscript 'CStone: A de novo transcriptome assembler for short-read data that identifies non-chimeric contigs based on underlying graph structure.' has been provisionally accepted for publication in PLOS Computational Biology.

Best regards,

Mihaela Pertea

Software Editor

PLOS Computational Biology

Mihaela Pertea

Software Editor

PLOS Computational Biology

---

## [Editor Report · Acceptance letter]

17 Nov 2021

PCOMPBIOL-D-21-01174R2 

CStone: A de novo transcriptome assembler for short-read data that identifies non-chimeric contigs based on underlying graph structure.

Dear Dr Archer,

I am pleased to inform you that your manuscript has been formally accepted for publication in PLOS Computational Biology. Your manuscript is now with our production department and you will be notified of the publication date in due course.

With kind regards,

Zsofia Freund
